# Improvement of Fire Resistance and Mechanical Properties of Glass Fiber Reinforced Plastic (GFRP) Composite Prepared from Combination of Active Nano Filler of Modified Pumice and Commercial Active Fillers

**DOI:** 10.3390/polym15010051

**Published:** 2022-12-23

**Authors:** Andry Rakhman, Kuncoro Diharjo, Wijang Wisnu Raharjo, Venty Suryanti, Sunarto Kaleg

**Affiliations:** 1Mechanical Engineering Department, Engineering Faculty, Universitas Sebelas Maret, Surakarta 57126, Indonesia; 2Department of Chemistry, Faculty of Mathematics and Natural Sciences, Universitas Sebelas Maret, Surakarta 57126, Indonesia; 3Research Center for Transportation Technology, National Research and Innovation Agency, Tangerang Selatan 15314, Indonesia

**Keywords:** composite, glass fiber reinforced plastic, nano filler, pumice

## Abstract

Glass fiber reinforced plastic (GFRP) composites have great potential to replace metal components in vehicles by maintaining their mechanical properties and improving fire resistance. Ease of form, anti-corrosion, lightweight, fast production cycle, durability and high strength-to-weight ratio are the advantages of GFRP compared to conventional materials. The transition to the use of plastic materials can be performed by increasing their mechanical, thermal and fire resistance properties. This research aims to improve the fire resistance of GFRP composite and maintain its strength by a combination of pumice-based active nano filler and commercial active filler. The nano active filler of pumice particle (nAFPP) was obtained by the sol–gel method. Aluminum trihydroxide (ATH), sodium silicate (SS) and boric acid (BA) were commercial active fillers that were used in this study. The GFRP composite was prepared by a combination of woven roving (WR) and chopped strand mat (CSM) glass fibers with an unsaturated polyester matrix. The composite specimens were produced using a press mold method for controlling the thickness of specimens. Composites were tested with a burning test apparatus, flexural bending machine and Izod impact tester. Composites were also analyzed by SEM, TGA, DSC, FT-IR spectroscopy and macro photographs. The addition of nAFPP and reducing the amount of ATH increased ignition time significantly and decreased the burning rate of specimens. The higher content of nAFPP significantly increased the flexural and impact strength. TGA analysis shows that higher ATH content had a good contribution to reducing specimen weight loss. It is also strengthened by the lower exothermic of the specimen with higher ATH content. The use of SS and BA inhibited combustion by forming charcoal or protective film; however, excessive use of them produced porosity and lowered mechanical properties.

## 1. Introduction

Recently, transportation industries have replaced metals with plastic materials. Plastics have a faster production cycle than that metal processing, so they save costs and increase product quality [1]. The limitation of weight is the main consideration in this material due to its high efficiency and economic value. The use of plastics in various modes of transportation is much better [2]. Plastics are commonly used in industry in the form of composites.

In producing composite material, it is necessary to add fiber as reinforcement. Glass fibers improve the mechanical properties and fire resistance of composites [3,4]. They have good insulation properties, high melting point, resistance to chemical and water environments, and fire resistance up to 1050 °C. It cannot be ignited if the heat flux is below 100 to 125 kW/m^2^ [5,6]. Glass fiber reinforced plastic (GFRP) composite with unsaturated polyester resins (UPRs) matrix has a big trend in the transportation industry.

UPRs are widely used as matrix composite materials in construction, transportation and other industries. This matrix has many advantages, such as good mechanical properties, corrosion resistance, good processability, low viscosity and density, high strength-to-weight ratio and fast production cycle [7,8,9]. However, the polymer matrix has low flame retardant properties and reaction to fire [10]. The flammability of polymers can be reduced by adding flame retardants [11]. The development of halogen-free flame retardants must be fabricated to meet the regulations and safety standards [12,13] of the government’s program towards green products [14].

An active inorganic flame retardant, aluminum trihydroxide (ATH), is the most popular flame retardant that is used in polyester composites [15]. ATH is relatively cheap, easy to apply in polymers and environmentally friendly in smoke density and low toxicity [16]. The presence of ATH produces good flame retardant properties and thermal stability of composite [17,18]. At a temperature of 80–220 °C, ATH decomposes into alumina and water vapor and forms thermal insulating charcoal [19]. Alumina forms a protective layer and moisture to lower the temperature of the system and acts as a diluent agent in the gas phase [15]. The addition of sodium silicate (SS) contributes to the formation of polysilicate during composite manufacturing [20]. Char forms larger because the polysilicate forms a protective layer and insulates the composite surface after exposure to heat, resulting in good product and fire-retardant features in the composite [21,22]. Boric acid (BA) has a good flame retardant and active insulating effect on polyester composites. During the combustion process, BA reduces heat because it produces wet B_2_O_3_ and volatile H_2_O. B_2_O_3_ forms charcoal between the combustion process and oxygen [23,24]. Smoke density also decreased with the addition of BA into the UPRs [25].

In addition to being influenced by the glass transition temperature (Tg) and decomposition temperature (Td) [7], composite properties are also influenced by the shape and size of the particles [26]. Nano-sized materials are very promising in improving the properties of polymer composites [10]. The addition of nanosilica can form hydrogen bonds between the silanol group on the nano surface and the carbonyl ester group on the soft segment, thereby improving the thermal properties and adhesion of polyester [27]. Certain compositions improve the mechanical properties of composites [28]. Nanosilica or silicon dioxide (SiO_2_) is commonly found in the pumice [29,30], kaolin [31] and clay [32]. Pumice is formed as a result of the rapid cooling of gases and volcanic material from volcanic eruptions [33]. Indonesia has active volcanoes, which are about 30% of the world’s active volcanoes in the world [34]. The pumice contains major elements of 58.3% SiO_2_, 12.4% Fe_2_O_3_ and 12% Al_2_O_3_ [35]. The characteristics of silica are influenced by the method and parameters of the synthesis process, which can be performed by acid or alkaline treatment [29].

The flame retardant properties of composites increased with the addition of ATH into polyester, which was marked by a decrease in the burning rate [36]. There is an increase in flame retardant, thermal stability and decomposition temperature of the composite by adding BA to the composite [37]. The addition of SS in the composite can result in better flame retardant properties and thermal stability [38]. Likewise, the incorporation of ATH, BA and SS into the polyester is able to overcome the flammability of the composite. However, the mechanical properties of the composites decreased along with the increase in the filler content of ATH, BA and SS [39]. Mechanical properties increased with the addition of polyester nanosilica composites [40]. The data of both properties can also be used to design some components or products according to the operating conditions.

In previous studies, many researchers synthesized pumice [29,30,35] and combined it with other flame retardants, such as phosphorus-based flame retardant, ammonium polyphosphate, zinc hydroxystannate, magnesium hydrate, ATH, nanosilica, nanoalumina and nanoclay [41]. However, at present, there are still few, and no one has even discussed the combination of four flame retardants at the same time. A combination of nano active filler particle pumice (nAFPP), ATH, SS and BA is researched to meet the fire resistance and mechanical properties of GFRP composites. The physical properties (TGA, DSC, SEM, FTIR, macro photo) were observed to give analysis contribution of both properties. All flame retardants (nAFPP, ATH, SS and BA) are analyzed and discussed to explore their contribution to fire resistance and mechanical properties. The purpose of this study was to expand knowledge about the synthesis of pumice by developing the sol–gel method and the performance of GFRP composites on fire resistance and mechanical properties. For this purpose, SEM morphological analysis was to determine the particle size of the synthesis results and fuel resistance test to determine the ignition time and the rate of combustion of the composite. Then, the analysis of flexural strength and impact strength to determine the mechanical properties of the composite.

## 2. Materials and Methods

### 2.1. Materials

Pumice was obtained from Rinjani Mountain, Lombok Island, West Nusa Tenggara, Indonesia. Chemicals were supplied by e-Merck. Unsaturated polyester resin 268 BQTN and methyl ethyl ketone peroxide catalyst were supplied by Singapore Highpolymer Chemical Products (SHCP). The chopped strand mat (EMC200) and woven rowing of E-glass fibers were supplied by PT. Makmur Fantawijaya Chemical Industries, Jakarta, Indonesia.

### 2.2. Methods

#### 2.2.1. Synthesis of Nano Active Filler Particle Pumice (nAFPP)

Pumice contains major elements of 58.3% SiO_2_, 12.4% Fe_2_O_3_, 12% Al_2_O_3_ [42]. The nAFPP was successfully synthesized using the sol–gel precipitation method. The pumice was washed with water and dried at 100 °C for 12 h. Pumice particles were produced by crushing and sieving with a size of 200 mesh (≤74 µm). They were washed and thermally activated at 680 °C for 1 h. Activated particles (100 g) were dissolved in 1000 mL of 2.5 M HCl and stirred at 300 rpm and 95 °C for 2 h. The mixture was filtered to obtain silica-rich pumice particles. Particles were washed with distilled water to reach pH 7. Sodium silicate solution was prepared by dissolving 10% wt of silica-rich pumice particles in 2 M NaOH at 95 °C for 2 h and stirrer at 300 rpm. The solution was filtered to separate residues and impurities [32]. The filtered solution was precipitated using 10 mL ethanol dispersant and 5 M HNO_3_ at 65 °C and pH 7. The solution was further filtered to obtain silica gel and washed with hot distilled water to remove impurities. Finally, the silica gel was dried at 80 °C for 4 h (Figure 1) [43].

#### 2.2.2. Preparation of Composites

Glass fiber reinforced plastic (GFRP) composites were made by a combination of hand layup press mold methods. Firstly, UPRs were mixed with fillers according to Table 1, and the mixture was stirred at 3000 rpm for 5 min, then it was allowed to stand so that the air bubbles disappeared [44]. The catalyst was added to the mixture at 1% by weight of the UPRs and stirred again so that the catalyst was well mixed. The fiber was cut to the size of the molding, and then the resin–filler mixture was poured into the molding according to the number of layers made. After 24 h, the composite was removed from the molding and then post-cured in an oven at 100 °C for 60 min [41]. The composites were cut to produce the test samples according to ASTM standards, i.e., ASTM D 635 for burning testing, ASTM D 790 for flexural testing and ASTM D 5941 for Izod impact testing [45,46].

#### 2.2.3. Testing of Composites

Testing of fire resistance refers to ASTM D635-03 with standard bar specimens of 125 ± 5 mm in length, 13.0 ± 0.5 mm in width, and 3.0 (−0.0 to +0.2) mm in thickness [47]. The specimen was clamped in a horizontal direction with 5 mm of clamping length. The specimen was marked in lengths of 25 mm and 75 mm, as shown in Figure 2 [47].

In this study, the flexural test was carried out with the ASTM D-790 standard using the Universal Testing Machine JTM for a three-point bending method at a crosshead speed of 2 mm/min with a load cell of 200 kg. The load was placed in the center of the specimen, which was supported by two supports. This testing resulted in the curve of load–deflection, and then the flexural strength was obtained [46]. In the test of three-point bending, the failure occurred when the specimen changed shape due to the loading of the specimen at 200 kg. The impact strength of Izod was determined using an impact testing machine following the ASTM D-256 standard.

SEM observations were performed using a scanning electron microscope JEOL JCM-7000. Observations were made on the fracture side of the composite specimen as a result of impact testing. Thermogravimetric analysis (TGA) was carried out on STA PT 1600 (TG-DSC) equipment. Samples were loaded in alumina pans and heated at a rate of 20 °C/min from 30 to 600 °C under a dry air atmosphere. The Fourier transform infrared (FTIR) spectra were recorded using an IR Prestige-21 Shimadzu at 400–4000 cm^−1^.

## 3. Results and Discussion

Figure 3 shows the morphology and particle size of the synthesis of pumice particles. SEM observations show that the silica particles have an average size of 30 ± 16 nm. This indicates that the dispersant will be adsorbed onto the surface of the silica particles and form a protective layer of macromolecules. This macromolecular protective layer will inhibit the growth of particles to obtain a smaller particle size with better dispersibility in the sol–gel deposition process [48,49,50].

Figure 4 shows the burning test results of the composites. The ignition time and burning rate of the C81, C82, C83 and C84 composites have similar properties. The composites containing nAFPP (reduction in ATH) have higher fire resistance compared to that without nAFPP, and the higher content of ATH until 4% gives a significant contribution to the increase in ignition time and reduced burning rate. The ATH releases water vapor, which functions as a diluent for volatile gases due to endothermic reactions when burned. ATH also forms an alumina layer (Al_2_O_3_), which functions as a protective layer from heat. This ATH reacts at a temperature of 180–200 °C into aluminum oxide through an endothermic reaction, which causes the UPRs to cool down, thereby reducing the pyrolysis product [39,51]. The cross-linking of UPRs also keeps the polymer chains from breaking under the influence of heat [52,53].

Both ignition time and burning rate show the best properties on the C81 for the composite containing 4% ATH and 1% nAFPP. Otherwise, the composite C84 with 4% nAFPP and 1% ATH has a higher burning rate and lower ignition time. This shows that the ATH gives a better contribution to the increase in fire resistance. All these fire retardants (nAFPP, ATH, SS and BA) are effective in increasing the fire resistance of the GFRP composite.

With the addition of BA, the protective layer is also stronger because BA melts at a temperature of 236 °C, and it dehydrates when heated above 300 °C [54]. If the heating continues, the boron oxide forms a protective film to inhibit the flame. Similarly, the presence of SS filler containing silicates can also increase the inhibition of flame. When SS mixes with UPRs, the silicates produce composites that have high thermal and mechanical properties [38,55] because SS forms an intumescent layer during the combustion process, so the temperature drops [56].

The nAFPP, which has a high silica content, causes the formation of a barrier preventing volatile evolution during degradation and increases the amount of char produced. In addition, nAFPP, which has a nano size, can reduce defects occurring on the GFRP so that it increases the mechanical strength of the composite [26].

Figure 5 shows the results of macro photos of the burnt composite sample surface. White spots on the surface of the composite are the remnants of unburned filler. Samples C81 and C82 show more spots compared to the other samples. The nAFPP content in this sample reached 4% by weight. This indicates that the filler is not well dispersed, caused by the stirring process during the manufacturing process of the composite. This could also be caused by agglomeration in the composites [17].

Figure 6 shows that the specimen experiences a deflection during loading. The specimen fractured at peak loading up to 103 N within 35 s, and the sample completely fractured within 63 s. Thus, specimen failure occurs in less than 60 s [57].

The flexural properties of GFRP composite specimens with various fillers nAFPP, ATH, SS and BA were tested, and the results are shown in Figure 7. It can be seen that increasing the nAFPP content up to 2% (wt) and reducing the ATH content up to 3% (wt) decreases the strength of the composite compared to the composite without filler (C). A further increase in the nAFPP filler content by up to 4% (reducing the ATH content up to 1%) could increase the composite strength compared to the unfilled composite. Specimen of C84 specimen has the highest flexural strength value of 58.5 MPa. It can be concluded that the addition of 4% of nAFPP increases the load-bearing capacity of the composite [58].

The relationship between nAFPP and commercial active fillers (ATH, BA, SS) with the impact strength of the composite is shown in Figure 8. The GFRP composite without filler has an impact strength of 69.22 KJ/m^2^. When the nAFPP content increases from 1 to 4% in weight, the impact strength of the composite also increases to 79.86, 85.99, 89.18 and 93.38 KJ/m^2^. The increase in impact strength was accompanied by an increase in nAFPP content up to 4% wt. The presence of nAFPP disperses in the matrix and easy to makes plastic deformation. Therefore, during the fracture of the composite in which nAFPP is well dispersed, the stress must be greater to initiate microcracking in the UPRs matrix. The impact energy will be mostly absorbed by the plastic deformation and more easily occurs in the vicinity of the nAFPP. Good nAFPP dispersion resulted in less agglomeration leading to better impact strength of the composite [59].

Figure 9 shows the result of SEM observations on the fracture area of impact test specimens. The control sample (C.a and C.b) shows that the composite without filler produces a lower interfacial bond, thereby reducing the impact strength is occurred of the composite. Meanwhile, the C84.a and C84.b samples show a strong interfacial bond between the fiber and the matrix due to the high amount of nAFPP can reduce the occurrence of voids. The composition produces a solid material that can support more loads [60] so that it has a high impact strength.

The TGA result in Figure 10 shows that the oxidative thermal degradation composites take place in two main stages. The first stage of degradation is occurred at about 200–300 °C, and the other takes place at or above 300 °C. The second stage comes to pass depolymerization, and the breakdown of the UPRs chain takes place at or above 300 °C [55]. Sample C81 has a lower initial temperature and a higher rate of weight loss from thermal decomposition compared to the others. Both BA and ATH have lower dehydration temperatures [61]. The results obtained in this study, the materials tested used a constant heating rate without any variation in the heating rate.

In Figure 11, the initial test curve shows a slow endothermic reaction depicted by a gentle descending curve. At temperatures less than 200 °C, the absorbed heat is used to evaporate water vapor and evaporation due to the initial decomposition process [62]. When the specimen is above 200 °C, an endothermic peak appeared at 282 °C for the C81 and C82 composites. This provides information that the composite with filler requires more heat than the composite without filler. The results of the combustion test also show that the C81 and C82 composites have a longer initial ignition time than the composites without filler (C). This endothermic reaction describes the beginning of the thermal decomposition of the composite, which produces volatiles. Substances that are volatile will be broken down into smaller fractions so that they are more easily oxidized [63].

In Figure 11, the curve starts to rise at 320 °C, and exothermic peaks occur at 346 °C for the C composite, 352 °C for the C81 and C82 composite, and 350 °C for the C83 and C84 composites. This exothermic reaction describes the thermal oxidation of volatiles with oxygen that produces heat (exothermic). The exothermic peaks of the three composites were at a higher temperature when compared to the C composite (without filler). This is because the filler prevents oxidation and is followed by an endothermic reaction during the decomposition of the filler. Filler decomposition produces water vapor, which plays a role in reducing the burning rate. The water vapor escapes in the flame and dissolves the concentration of combustible gases from the polymer matrix. This condition causes oxygen access on the composite surface to be limited [64].

Figure 12 shows the FTIR spectra curve in the range of 400–4000 cm^−1^. Absorption -H (3476 cm^−1^), aromatic = strain C-H (3028 cm^−1^), aliphatic vibration C-H (2942 cm^−1^), aromatic ring (699–857 cm^−1^) and O = C-O (1732 cm^−1^) was detected for composite C (without filler), which is a product of UPR gas decomposition [65]. In the other composites, peaks at 1067–1063 cm^−1^ were detected, which defined the strain asymmetry of Si-O-Si [66]. This indicates the presence of silicate compounds derived from AFPP and SS fillers. The intense band at 633 cm^−1^ for the C81 composite, 616 cm^−1^ for the C82 composite and 616 cm^−1^ for the C83 composite is the vibrational strain of Al-O. This is because the four composites contain ATH filler containing Al, except for the C84 composite because the ATH filler content is very low [67].

The observed bands at 1279 and 1124 cm^−1^ for C81, 1280 and 1123 cm^−1^ for C82, 1280 and 1121 cm^−1^ for C83, and 1282–1121 cm^−1^ for C84 were designated as vibrations of B-O-B. In this vibration, the peak value is not much different because the composite composition contains the same BA filler. In the four composites that were added with filler, OH peaks also appeared at 3453 cm^−1^, 3512 cm^−1^, 3449 cm^−1^ and 3446 cm^−1^, which would dilute the volatiles so that oxidation was reduced.

## 4. Conclusions

The fire resistance of the GFRP composites increases along with the addition of ATH (reduction nAFPP), shown by the increase in ignition time and the decrease in the burning rate. Otherwise, the composites with higher nAFPP (lower ATH) have higher bending and impact strength. The ATH and nAFPP are more effective in inhibiting the flame and increasing the strength, respectively. BA is a good insulator to inhibit the flame by producing a protective layer in different temperature stages of dehydration, while SS produces a protective layer that does not melt easily. Both BA and SS in higher content can decrease the GFRP composite. The composite can be optimized by arranging the optimum combination content of each filler according to its operating conditions. The fracture of the composite shows a strong interfacial bond between fiber and matrix due to the high amount of nAFPP that can reduce the occurrence of voids. The composite with filler needs more heat for the decomposition process compared to those without filler.

## Figures and Tables

**Figure 1 polymers-15-00051-f001:**
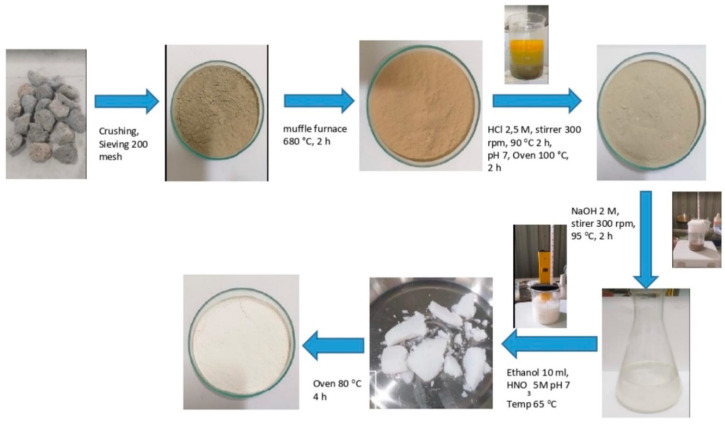
Synthesis process of nano active filler pumice particle.

**Figure 2 polymers-15-00051-f002:**
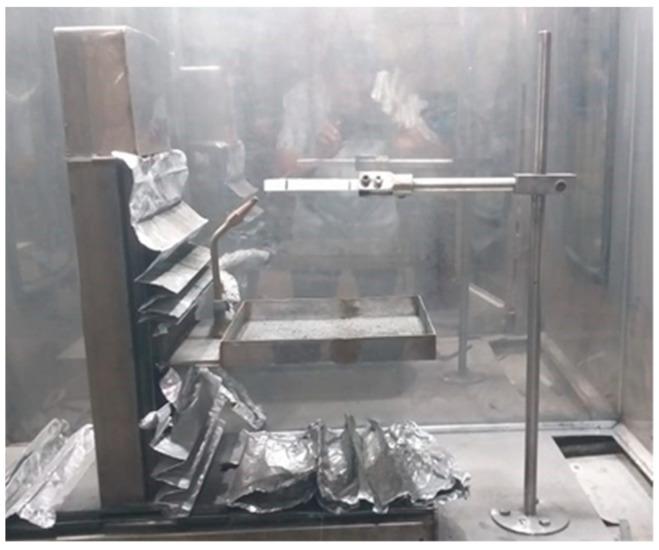
Burning test. Reprinted with permission from Ref. [41]. 2015, Diharjo, K.

**Figure 3 polymers-15-00051-f003:**
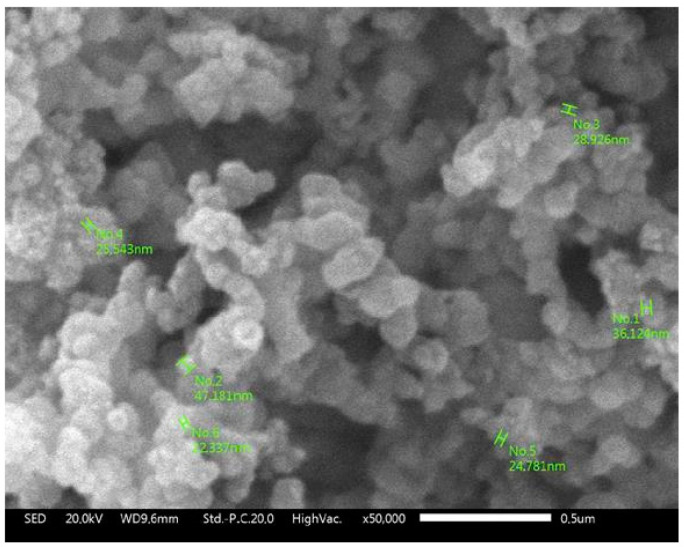
The particle size distribution of nAFPP was measured by SEM.

**Figure 4 polymers-15-00051-f004:**
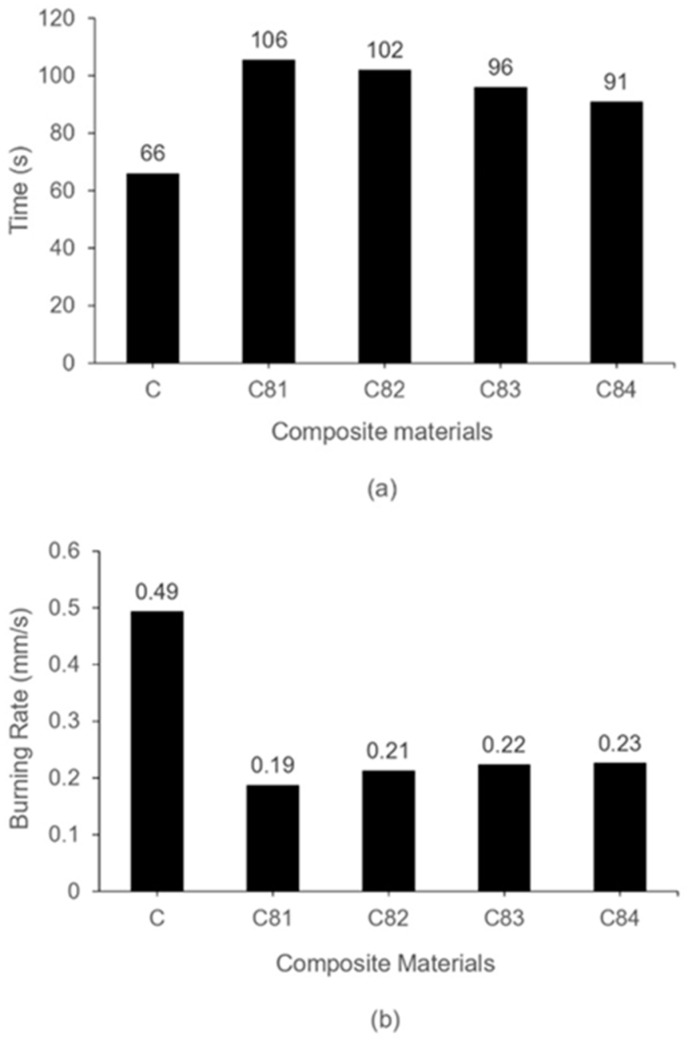
Burning test of composites (**a**) ignition time (**b**) burning rate.

**Figure 5 polymers-15-00051-f005:**
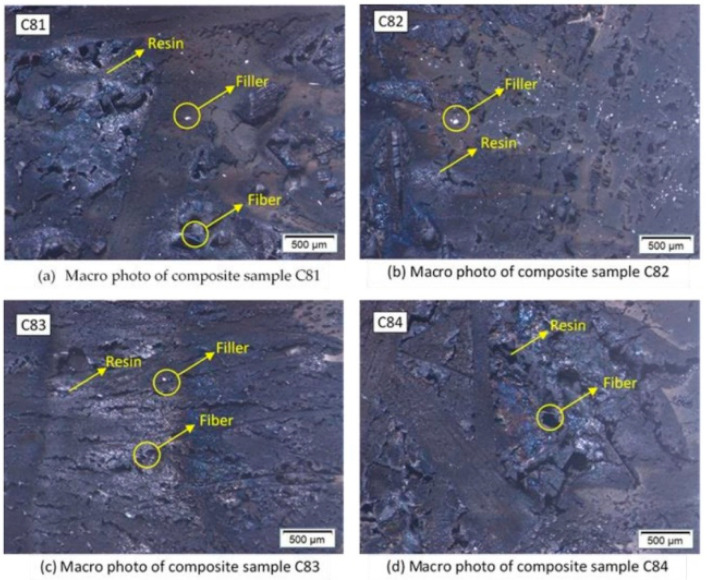
Macro photo on the burnt surface of composites.

**Figure 6 polymers-15-00051-f006:**
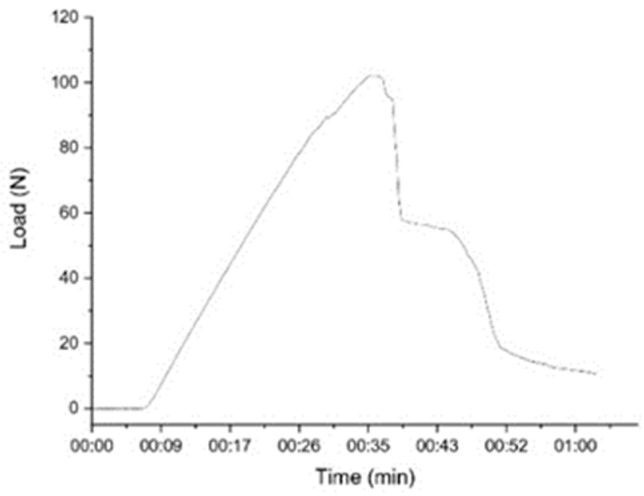
Flexural test according to fracture time.

**Figure 7 polymers-15-00051-f007:**
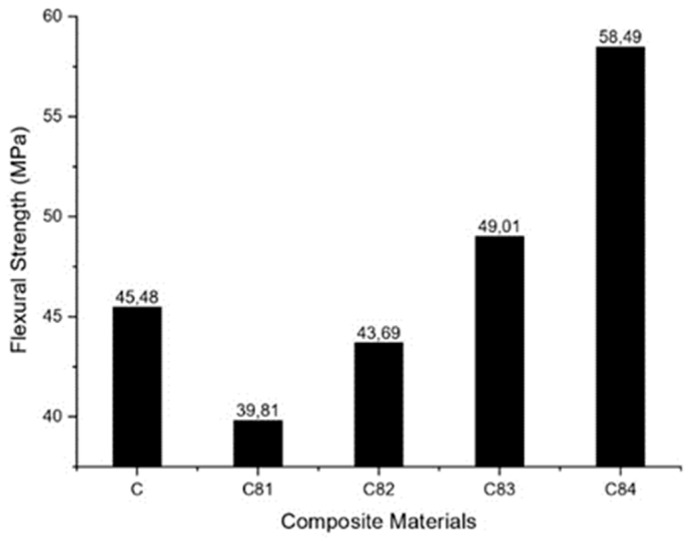
Flexural strength of composites.

**Figure 8 polymers-15-00051-f008:**
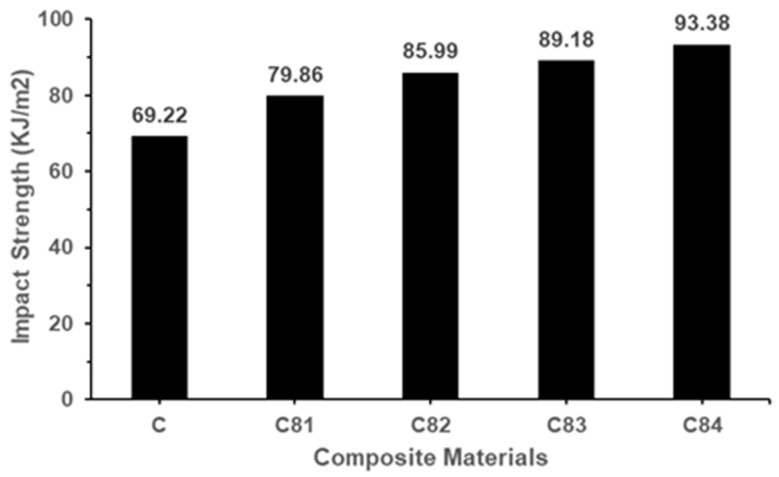
Impact strength of composites.

**Figure 9 polymers-15-00051-f009:**
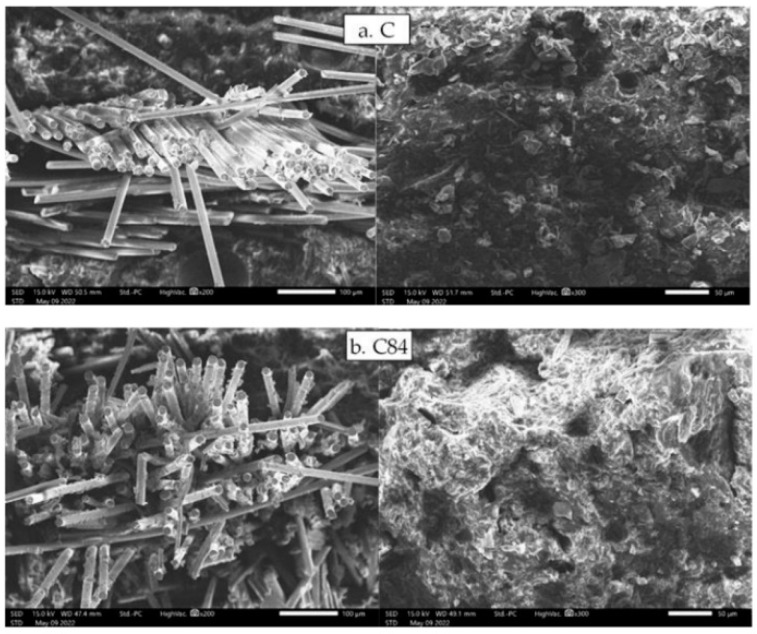
SEM observation on impact testing specimens on impact testing specimens: (**a**) GFRP composite without filler and (**b**) GFRP composite with filler.

**Figure 10 polymers-15-00051-f010:**
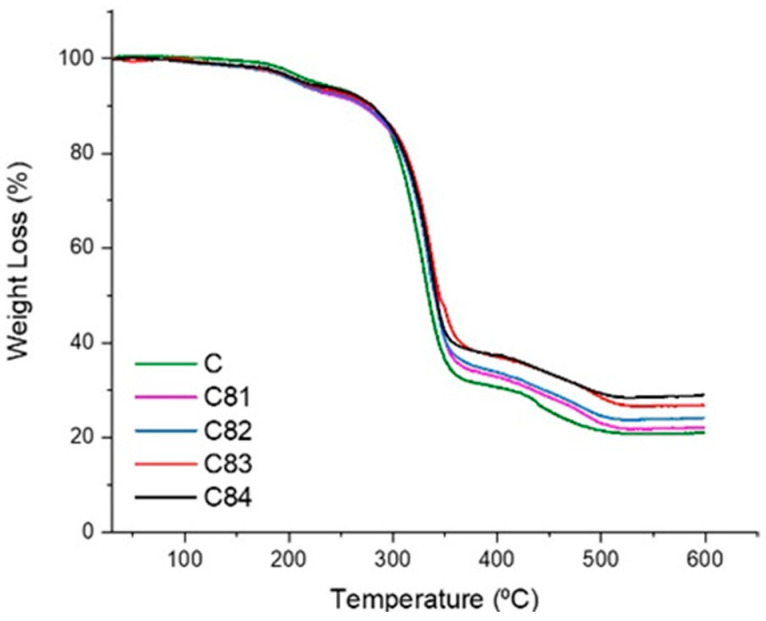
TGA Thermograms analysis of composites.

**Figure 11 polymers-15-00051-f011:**
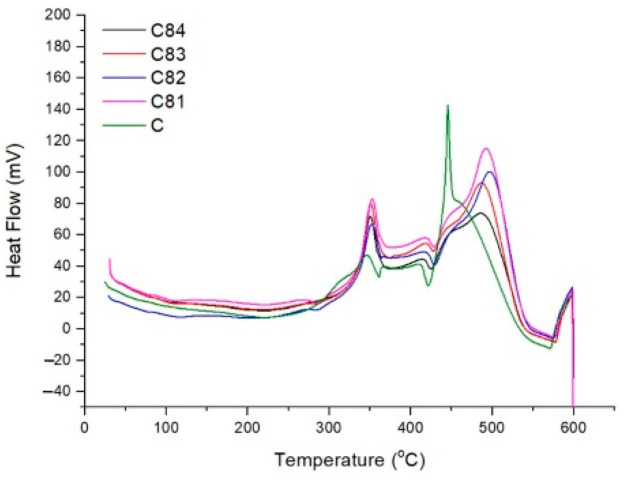
Differential scanning calorimetry (DSC) analysis of composites.

**Figure 12 polymers-15-00051-f012:**
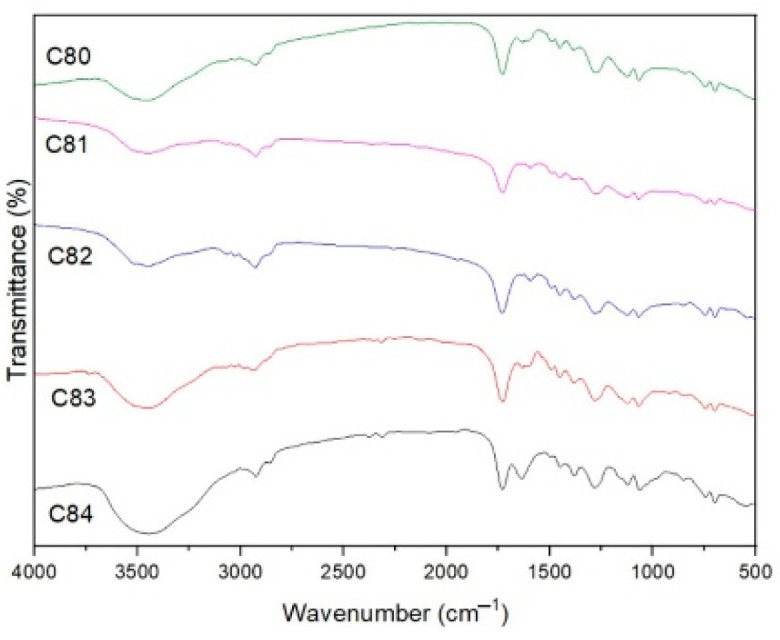
FTIR spectra of composites.

**Table 1 polymers-15-00051-t001:** Composite composition of GFRP with nAFPP, ATH, SS and BA.

No	Composite Code	GF	UPRs	Filler			
				nAFPP	ATH	SS	BA
		wt%	wt%	wt%	wt%	wt%	wt%
1	C	20	80	0	0	0	0
2	C81	20	70	1	4	2	3
3	C82	20	70	2	3	2	3
4	C83	20	70	3	2	2	3
5	C84	20	70	4	1	2	3

## Data Availability

The data presented in this article are available from the corresponding author.

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
