# Peer review of "Improvement of Fire Resistance and Mechanical Properties of Glass Fiber Reinforced Plastic (GFRP) Composite Prepared from Combination of Active Nano Filler of Modified Pumice and Commercial Active Fillers"

_polymers, 2022, doi:10.3390/polym15010051_

Round 1
Reviewer 1 Report
This study is a "good one" from the point of view of originality and scientific level which is deserving attention. This study will be a good one for other groups working on the similar matter of subjects, to be able to carry out similar and possibly further studies in the future. My main comment/criticism is that the authors do not consider the importance of the glass transition temperature of the GFRP material on its fire behaviour. I consider this manuscript suitable for publication after the following comments have been addressed:
1) Lines 1 to 2, “GFRP composites have great potential to replace metal components in vehicles by improving the fire resistance while maintaining its mechanical properties.” The reviewer suggests being careful about saying that GFRP materials have good fire resistance. GFRP material often requires protective measures, due to the flammability, the heat and smoke release and, more importantly for structural applications, the loss of mechanical properties for relatively low temperatures.
2) In the introduction section, the authors failed to acknowledge a key parameter related to the fire behaviour of GFRP material: the glass transition temperature. It’s well known that the mechanical behaviour of GFRP at high temperatures is strongly dependent on this parameter. The authors must recognize this aspect within the text. Regarding the Tg of the material, please remind that the manufacturing process also have influence on this parameter. Please find below some reference that can be useful regarding this aspect:
Mazzuca et al. DOI: https://doi.org/10.1016/j.conbuildmat.2022.128340, Tg of GFRP material produced by vacuum infusion
Rosa et al. DOI: http://dx.doi.org/10.2749/guimaraes.2019.0861, Tg GFRP material produced by pultrusion
Chodhury et al. DOI: 10.1007/s10694-009-0116-6, Tg of GFRP material produced by hand layup
3) At the end of the introduction section, the aim of study should be more emphasized.
4) The fibre-to-weight ratio of the material tested in this study should be specified in section 2. The fire behaviour of the composite can also depend on this parameter.
5) In the 3 points bending tests, how was the time to failure? Please specify
6) The authors should specify the capacity of the load-cell used in the mechanical tests. In addition, it is not clear how the “fire load” (200 kg) was defined?
7) The authors should give more details about the test procedure followed in the flexural tests.
8) Lines 156 to 158, How the heating rate was defined in the TGA tests? Note that this parameter has a great influence on the determination of the decomposition temperature. In addition, if feasible, the reviewer also suggests performing tests in nitrogen atmosphere.
9) Figure 4, Please check that the figure does not exceed the page’s limit.
10) It is not clear how the flexural strength was defined. It would be also useful some information about the failure modes.
11) In the conclusions, the authors should explain the significance and shortcomings of the research work, instead of repeating the results obtained before.
12) Please note that the cited references are quite old. The authors should include the research work about the fire resistance of GFRP materials developed in the past 2-3 years.
Author Response
Thanks for the reviewer comment for improving my paper and the revision has been done.

Reviewer 2 Report
The paper will be ready for publication after major revision.
This work is original, novel and important to the field.
Revise the paper according to the attached pdf file.

Author Response
Thank you very much for the reviewer comments to improve my paper, Thank you

Round 2
Reviewer 1 Report
Although the authors did an effort to improve the quality of the paper, they did not address most of the comments raised by the reviewers, there are still several aspects that need to be clarified/discussed in more details. For the next round, please use track changes, in order to easily detect what the authors added/removed.
1) The reply that the authors gave to response 1 is very cursory and must be improved. The authors should highlight the main advantages of FRPs over traditional materials, for instance high strength-to-weight ratio, lightness, ease of handling and durability. It’s not just a matter of “increasing the material properties”!
2) The authors completely failed to acknowledge the importance of the glass transition temperature, they only report in the introduction this sentence: “Composite properties are also affected by the glass transition temperature (Tg)”, which again, is very cursory. As suggested in the previous version, the authors should provide more details about the key correlation between the Tg and the mechanical behaviour at elevated temperature of GFRP materials, highlighting also the scatter of this property when testing material produced with different manufacturing methods. Unlike stated in point 2, none of the suggested references were used to improve this section.
- https://doi.org/10.1016/j.conbuildmat.2022.128340
- Rosa IC, Firmo JP, Correia JR, Mazzuca P. Influence of elevated temperatures on the bond behaviour of GFRP bars to concrete – pull-out tests, IABSE Symposium Guimarães, Portugal (2019).
- https://link.springer.com/article/10.1007/s10694-009-0116-6
3) The authors failed to acknowledge the reviewer suggestion. In this new version, they only add information about the tests carried out without highlighting the novelty and gap they want to fill with this work. Please improve this aspect.
4) The reply to point 5 is not scientifically correct. The authors should specify in minutes the time to failure. As the authors know, when testing FRP at high temperatures, it is suggested to have failure within 10 minutes to avoid short term creep effects.
5) The reply to point 7 is also not clear. The static and cinematic aspects of a beam loaded in 3-point bending is well know and does not deserve to be included in the manuscript. The authors should specify if the flexural properties correspond to residual properties (so after heating the specimens the load was applied at ambient temperature) or if the load was applied at constant temperature.
6) Again, the authors did not provide information about the definition of the heating rate in the TGA tests. Could you please provide additional information? Moreover, in the test, the authors should acknowledge the readers that the obtained results are only related to materials tested with the heating rate used in this study. TGA results would vary if using higher or lower heating rate!
Author Response
Thank you for your good comments in the revision

Reviewer 2 Report
The new comments on that paper are listed in the attached pdf file.
Author Response
Thank you very much for the comment to improve my article

Round 3
Reviewer 1 Report
Analyzing the authors' answers, I found that the authors did not respond concretely to all the mentioned comments. Therefore, following these findings, the decision remains the same with the mention of responding promptly and concisely to previously unresolved comments.
1. Reply to Response 1; again, the is some information which must be clarified. Low cost compared to traditional materials (concrete, timber or steel)? This is not true. In addition, a key problem of GFRP is not only the fire resistance but mostly the fire reaction properties. It is strongly suggested to revise those sentences.
2. Reply to Response 2, As suggested in the previous version, the authors should provide more details about the key correlation between the Tg and the mechanical behaviour at elevated temperature of GFRP materials, highlighting also the scatter of this property when testing material produced with different manufacturing methods. Please check the results reported in the studies suggested in the previous round.
3. Please note that the information reported by the authors regarding the degradation of the flexural properties when approaching the Tg does not correspond with what reported in the reference cited by the authors. Please note that there are certain mechanical properties (tensile ones) which are not affected by temperature lower than 300 C (so much higher than the Tg) since their mechanical response is more dependent on the thermos mechanical properties of the fibres rather than that of the matrix.
4. Reply to Response 4, The authors should absolutely acknowledge in the paper that failure occurred in less than 1 minutes. Usually, FRP should fail in an interval ranging from 1 to 10 mins. In addition, please check that 2 mm/s is the right value; it seems extremely high and non-adequate to the type of tests performed. In addition, please note that the English used by the authors is quite difficult to understand. Please, revise those sentences.
5. Reply to Response 5, I really don’t understand this reply. First, I still don’t understand if the tests were performed under residual or steady-state conditions. Then, in contrast with what reported in the reply to response 4, the authors say that tests were performed at a crosshead speed of 10 mm/minutes (in the previous reply they say it was 2 mm/s). Finally, saying that “failure occurs when the specimen changes shape due to loading on the specimen” is not scientifically correct. The authors should carefully revise those sentences.
6. Reply to Response 6, Again, the authors did not provide information about the definition of the heating rate in the TGA tests. Could you please provide additional information? Moreover, in the test, the authors should acknowledge the readers that the obtained results are only related to materials tested with the heating rate used in this study. TGA results would vary if using higher or lower heating rate!
Author Response
Thank you very much for your advice

Reviewer 2 Report
Accept.
Author Response
Thank you very much that you have agreed with my revision
